# One Bicopper Complex with Good Affinity to Nitrate for Highly Selective Electrocatalytic Nitrate Reduction to Ammonia

**Kang-Yu Zeng [1], Jun-Jie Wang [2],*, Xiang Fang [2] and Zuo-Xi Li [1],***

[1] Institute of Materials Science and Devices, School of Material Science and Engineering,
Suzhou University of Science and Technology, Suzhou 215009, China
[2] School of Chemistry and Chemical Engineering and Anyang Key Laboratory of New Functional Complex
Materials, Anyang Normal University, Anyang 455000, China
* Correspondence: jjwang@aynu.edu.cn (J.-J.W.); lizx@usts.edu.cn (Z.-X.L.)

**Abstract:** Ammonia ($NH_3$) plays an irreplaceable role in human life as a promising energy carrier and indispensable chemical raw material. Nitrate electroreduction to ammonium (NRA) not only removes nitrate pollutants, but also can be used for efficient $NH_3$ production under ambient conditions. However, achieving high efficiency and selectivity of electrocatalysts is still a great challenge. Herein, a complex $Cu_2(NO_3)_4(BMMB)\cdot H_2O$ with a bicopper core is assembled by $Cu(NO_3)_2\cdot 3H_2O$ and 1,4-bis{[2-(2′-pyridyl)benzimidazolyl]methyl}benzene (BMMB) for NRA under alkaline conditions. The optimal sample showed excellent nitrate reduction performance with the $NO_3^-$ conversion rate of 70%, Faradaic efficiency of up to 90%, and $NH_3$ selectivity of more than 95%. The high-catalytic activity is mainly due to the ingeniously designed copper cores with strong affinity for $NO_3^-$, which accelerates the transferring rate of adsorbed nitrate on the Cu surface and increases the efficiency of rate-determining step ($NO_3^- \rightarrow NO_2^-$) in the whole catalytic process. Therefore, the transformation of surface-exposed nitrate can be rapidly catalyzed by the Cu active sites, facilitating the conversion efficiency of nitrate.

**Keywords:** electrocatalysts; copper complex; nitrate affinity; nitrate electroreduction; ammonium production





## 1. Introduction

Currently, water pollution poses a major potential threat to public health and lives. Nitrate ($NO_3^-$) plays a non-negligible role among the pollutants [1]. In our daily life, the use of fertilizers, chemical production, and fossil fuels causes massive diffusion of nitrate and nitrogen oxides to surface water and groundwater, which resulted in the water quality not meeting the standard [2,3]. In addition, nitrate can be converted into nitrite ($NO_2^-$) in the human body, which causes severe health problems; for instance, gastric cancer and methemoglobin [4–6]. Furthermore, high concentrations of nitrate in the air can lead to acid rain, while in water, high concentrations can contribute to eutrophication [7].

To efficiently denitrify $NO_3^-$, numerous methods, such as reverse osmosis, adsorption, catalytic reduction, and biological denitrification have been used to treat $NO_3^-$ [8–11]. However, these techniques have shown some weaknesses. For example, reverse osmosis and adsorption need a post-treatment of highly concentrated $NO_3^-$, catalytic reduction is costly, biological denitrification is complicated and requires a long time to treat [12]. Electrocatalytic reduction method is more effective to treat nitrate than the above-mentioned methods. In addition, the highly selective reduction product, ammonia ($NH_3$), can be obtained by an external power supply at room temperature under atmospheric pressure [13–16]. The most desirable product of electrocatalytic nitrate reduction is $NH_3$, considering that $NH_3$ is an industrially important raw material and a promising liquefied fuel [17]. Industrial $NH_3$ is mainly a synthesis using Haber-Bosch process, which operates at high temperature (400~600 °C) and high pressure (>400 atm) [18]. However, $NO_3^-$ electrocatalytic reduction to valuable $NH_3$ is an environmentally-friendly, safe, and sustainable

process. According to the reported literatures, the rate-determining step of NRA is the conversion of $NO_3^- \rightarrow NO_2^-$; then, the important intermediate NO will be selectively converted into nitrogen or ammonia depending on electrode materials, solution PH, applied voltage, etc. [19–21]. The key point affecting the rate-determining step is the rate of nitrate adsorption on the catalyst surface. Therefore, the suitability of an electrocatalytic material for electrocatalytic nitrate reduction is determined by the ability to effectively adsorb $NO_3^-$ [22]. Among these methods, the NRA technique may be the most suitable for industrial production.

With regard to NRA, electrode materials mainly determine the electrocatalytic performances. At present, noble metal catalysts, such as gold (Au), rubidium (Ru), and platinum (Pt), have been shown to be excellent electrocatalytic materials [23–28]. Among these materials, although the performance of precious metals is excellent, the cost is expensive and there is no possibility of large-scale use. Transition metals are more suitable materials due to their low price and high electrocatalytic activity and stability. With regard to transition metal catalysts, copper (Cu) is one of the most desirable metals with high Faradaic efficiency (FE) and selectivity toward $NH_3$ [29,30]. Koper et al. demonstrated that Cu has the highest activity among transition metal catalysts [31]. Wu et al. reported that Cu nanosheets (Cu NSs) have high electric-double-layer-capacitance of 422 mF, indicating a very large electrocatalytic active area. Therefore, Cu NSs exhibit high ammonia selectivity of 99.7% nitrate removal rate [32].

Previous researches declare that some complexes show the ability to produce ammonia by nitrate electrocatalytic reduction. Jakubikowa et al. clarified that $[Co(DIM)]^{3+}$ (DIM = 2,3-dimethyl-1,4,8,11-tetraazacyclotetradeca-1,3-diene) has an active $NO_3^-$ electrocatalytic reduction to valuable $NH_3$, and density functional theory (DFT) was used to study the electronic structure and reaction mechanisms [33]. However, improving the NRA efficiency of complexes is still a challenge. Some efforts have been proposed to improve the catalytic activity. Luo et al. reported that Th-BPYDC (BPYDC = 2,2′-bipyridine-5,5′-dicarboxylic acid) has a low catalytic activity of NRA, but once supported by single-site copper catalyst, it presents 225 $\mu mol\,h^{-1}\,cm^{-2}$ yield and 92.5% Faradaic efficiency for NRA activated at 0.0 V (vs. RHE) [34]. Zhu et al. used an in-situ synthesis method to populate Cu nanoclusters in CuH-HTP (HHTP = 2,3,6,7,10,11-hexahydroxytriphenylene) [35]. In addition, Xu et al. used a similar approach to fill Cu nanoparticles in CeBDC ($H_2BDC$ = 1,4-benzenedicarboxylic acid) [36]. All of these methods are based on the loading of metal particles with a complex carrier, and the active center of catalysis is still the metal particle. Could the initial complexes have good catalytic activity?

With the above assumptions, we proposed a strategy for improving catalytic activity through designing a new complex $Cu_2(NO_3)_4(BMMB)\cdot H_2O$ (CuBMMB) with a bicopper core, which exhibits high activity for NRA. Specifically, at −0.53 V vs. reversible hydrogen electrode (RHE), CuBMMB achieves $NO_3^-$ conversion rate and Faradaic efficiency (FE) of up to 70 and 90%, and $NH_3$ selectivity of more than 95% in 1 M KOH solution with 200 ppm $NO_3^-$-N. The structure and composition are characterized in detail. To study the selectivity, mass activity, and durability, a combination of characterization tools and electrochemical tests was conducted. Finally, the mechanisms for the significant improvement in electrochemical performance are disclosed.

## 2. Results and Discussion

### 2.1. Synthesis of Consideration

First, the semirigid organic 1,4-bis{[2-(2′-pyridyl)benzimidazolyl]methyl}benzene (BMMB) chosen as the desirable ligand is attributed to the fixed bidentate bridging mode, and it makes the coordination with metal ions easier to be regulated (Figure S1). Second, the copper ion possesses good NRA performances. Third, it is important to increase the reaction rate in the rate-determining step ($NO_3^- \rightarrow NO_2^-$) of NRA process, and it is a practical strategy to focus on increasing the adsorption efficiency of nitrate on the surface of catalytically active centers. Therefore, we have used copper nitrate as the metal salt,

and successfully achieved good affinity of $Cu^{2+}$ ion with the nitrate ion, which led to the acceleration of the transferring rate of adsorbed nitrate on the Cu surface and improvement of the NRA performances.

### 2.2. Crystal Structure of CuBMMB

Single-crystal X-ray diffraction analysis reveals that CuBMMB crystallizes in the monoclinic space group $C2/c$ (see Supplementary Material). The asymmetric unit consists of one crystallographically independent $Cu^{2+}$ ion, two nitrate ions, one BMMB ligand, and one free water molecule. As shown in Figure 1, the $Cu^{2+}$ ion is coordinated by two BMMB nitrogen atoms and two nitrate oxygen atoms, which yield a square coordination environment. The Cu–N bond lengths are 1.952(3) and 1.978(3) Å, and the Cu–O bond lengths are 1.968(3) and 1.980(3) Å, which are all comparable to those typically observed values in the related copper complexes. Two terminal groups of BMMB connects one $Cu^{2+}$ ion, and thus each BMMB ligand connects two $Cu^{2+}$ ions. However, the two other coordination sites of $Cu^{2+}$ ion are occupied by the monodentate nitrate ion, which hinders the extension of metal node. Finally, the BMMB ligand connects the $Cu^{2+}$ ion into the complex molecule with a bicopper core.

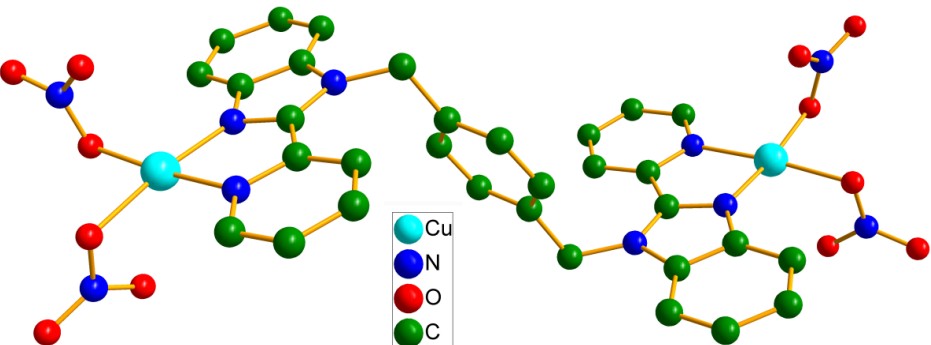

**Figure 1.** The coordination environment of $Cu^{II}$ ion and bicopper core in CuBMMB.

The X-ray diffraction (XRD) experimental pattern is well-matched with the simulated pattern (Figure 2a), proving the high purity and crystallinity of CuBMMB. With regard to CuBMMB, the distinct peaks at 9.53, 14.57, 18.58, 22.28, and 24.75° can be indexed to the $(2\,0\,\bar{2})$, $(4\,0\,0)$, $(6\,0\,\bar{2})$, $(3\,1\,2)$, and $(7\,1\,\bar{2})$ facets. The SEM images (Figures S1 and 2b,c) of CuBMMB present a loose structure with a large number of micron pores on the surface. From the magnified Figure 2b, it can be seen that CuBMMB particles are composed of many small rods with flocculent surfaces interlaced and woven. Not surprisingly, this open porous structure could accelerate the mass transfer of nitrate ions and products. SEM elemental mapping (Figure 2d) of CuBMMB shows that C, N, O, and Cu elements are homogeneously dispersed on the surface of CuBMMB. Energy dispersive X-ray spectroscopy (EDS) (Figure S2) calculates the content of each component and the results are shown in Table S1. The proportions of carbon, nitrogen, oxygen, and copper are 73.25, 4.43, 15.05, and 7.27%, respectively.

There are many identical functional groups in the FT-IR spectrum (Figure 3a) of BMMB and CuBMMB. The peak appearing at 1519 $cm^{-1}$ corresponds to the C=N stretching vibration in the benzimidazole ring, the peaks at 1477 and 3050 $cm^{-1}$ could be identified as the stretching vibration of the benzene ring skeleton [37]. Some peaks correspond to C=C bending modes (1450–1600 $cm^{-1}$), C–O stretching modes (1200–1250 $cm^{-1}$), and C–H stretching modes (680–880 $cm^{-1}$). In particular, the peak at 1350–1400 $cm^{-1}$ corresponds to the –N=C in BMMB. The peak at 1250–1300 $cm^{-1}$ corresponds to the N–O–Cu [38], which matches the results of the O 1s spectrum (Figure S3) of CuBMMB. When the CuBMMB powder was subjected to the TG experiment (Figure 3b), the crystalline water (3.8% observed, 4.0% calculated) was all lost in the region of 81–123 °C. The compounds are stable below 160 °C and decompose rapidly after 160 °C. At 401 °C, a large quantity of carbon

begins to be converted into $CO_2$. After almost complete decomposition of carbon at 640 °C, the main component of the final product was Cu−O with mass of 16.3% (17% calculated). XPS characterization of CuBMMB was carried out to investigate the composition and state of the complex. The XPS pattern for the complex displays the presence of C, N, O, and Cu elements (Figure 3c), which corresponds to the above-mentioned element mapping results. Cu 2p XPS spectra were shown in Figure 3d, and could be divided into three pairs of characteristic peaks of $Cu^{2+}$, $Cu^{+}$/Cu, and satellites [36,39]. A pair of peaks appearing at 934.39 and 954.25 eV accompanied by a pair of satellite peaks at 943 eV (Cu $2p_{3/2}$) and 963 eV (Cu $2p_{1/2}$) could be attributed to $Cu^{2+}$ [36]. Last pair of peaks at 932.42 eV (Cu $2p_{3/2}$) and 952.3 eV (Cu $2p_{1/2}$) were assigned to $Cu^{+}$/Cu [40,41]. Interestingly, there is a tendency for $Cu^{2+}$ to be reduced to $Cu^{+}$/Cu during the synthesis of the complexes, the ratio of $Cu^{2+}$ for CuBMMB (50.22%) is almost the same with $Cu^{+}$/Cu; this phenomenon is consistent with the study by Luo et al. [28]. Furthermore, the high magnification O 1s, C 1s, and N 1s spectra were presented in Figure S3. N 1s spectra are deconvoluted to three peaks of pyridinic-N (398.54 eV), pyrrolic-N (399.86 eV), and Cu−N (405.73 eV) [42], and the contents are 51.9, 30.45, and 17.65%, respectively. The O 1s spectra are fitted to four peaks of H−O (533.3 eV), Cu−O (Cu−O, 532 eV), N−O (532.7 eV), and C−O (531.1 eV) [43,44], and the contents are 11.09, 38.87, 40.98, and 9.06%, respectively; these analyzed bonds correspond to the results of the IR spectra. The relative percentage content regarding each component of multiple nitrogen and oxygen functionalities was listed in Table S1. Complexes containing N- and O- functional groups usually have better wettability and electrical conductivity, which is more conducive to electrocatalytic reactions.

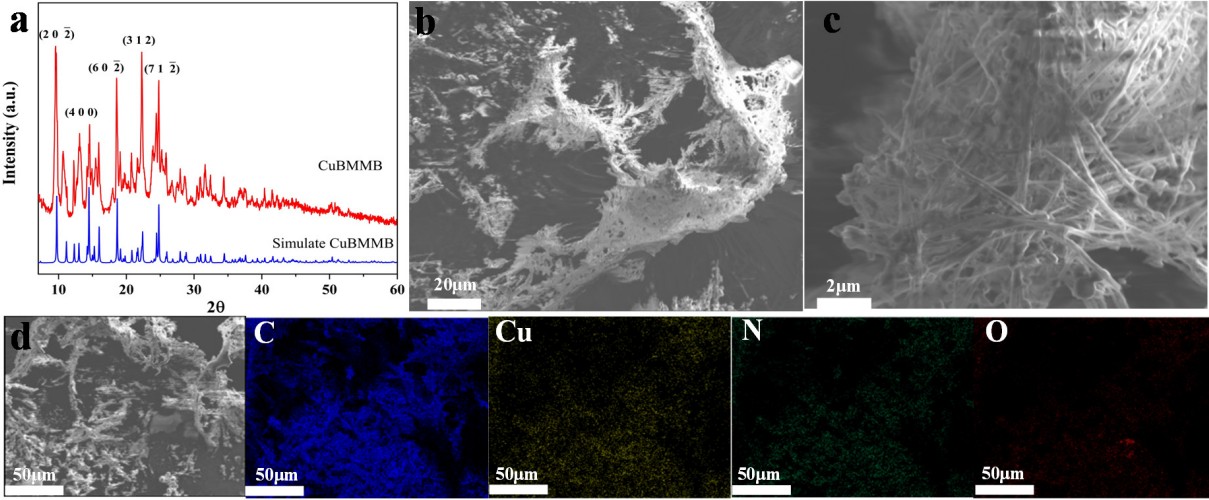

**Figure 2.** Structure, morphology, and composition analyses: (**a**) XRD pattern; (**b**,**c**) SEM image of CuBMMB; (**d**) SEM elemental mapping of C, Cu, N, and O in CuBMMB.

A series of electrochemical tests on the NRA properties of CuBMMB were carried out on 660E electrochemical workstation. Cyclic voltammograms (CV), linear sweep voltammetry (LSV), and chronoamperometry (I-t) curves were evaluated via a three-electrode configuration in an H-type electrolytic cell. The CuBMMB, saturated calomel electrode (SCE), and platinum (Pt) foil acted as working electrode, reference electrode, and counter electrode, respectively. The 1 M potassium hydroxide solution (KOH, 40 mL) was respectively added to the anode and cathode chambers, and $KNO_3$ (200 ppm $NO_3^{-}$-N) was added to the cathode chamber. The transformation of $NO_3^{-}$ to $NH_3$ was investigated over a reaction time of 5 h via the potentiostatic method with different potentials. The reductive products ($NO_3^{-}$-N, $NO_2^{-}$-N, $NH_4^{+}$-N) were determined by colorimetric methods, and the calibration curves were depicted in Figure S4. As shown in the Cu coordination environment (Figure 1), the coordinated $NO_3^{-}$ and the $NO_3^{-}$ on the surface of the electrode material could be converted to $NO_2^{-}$ with the electrochemical reaction. The

total nitrogen content in the solution could gradually increase with time, and reach 208.7 ppm after 5 h (Table S2). In view of this situation, we use the changed $NO_3^-$ concentration when calculating the $NO_3^-$ removal efficiency, Faradaic efficiency, and selectivity toward $NH_3$. As shown in Figure 4a, the LSV curves of CuBMMB showed a rapid increase in current density with the presence of $NO_3^-$, indicating that CuBMMB has the activity to catalyze the $NO_3^-$ reduction. Moreover, NRA performance of carbon cloth was investigated (Figures S5 and S6), and showed that carbon cloth exhibited significant lower efficiency in catalyzing the $NO_3^-$ reduction than CuBMMB in the potential window. Furthermore, the $NO_3^-$ conversion rate and FE were significantly lower than CuBMMB. Therefore, it can be concluded that the carbon cloth has little effect on the NRA performance of CuBMMB. The NRA performance of CuBMMB at different potentials ($-0.63$, $-0.53$, $-0.43$, and $-0.33$ V vs. RHE) was investigated. As displayed in Figure 4b, the nitrate removal efficiency gradually increased with the improvement in potential and achieved the maximum value of 89% at $-0.63$ V. However, Faradaic efficiency toward $NH_3$ production is lowest at 61.4%, which was mainly due to the competing HER, as it played a dominant role at high potential. At $-0.53$ V, Faradaic efficiency was around 90%, $NO_3^-$ removal efficiency was 73.4%, $NH_4^+$ selectivity and ammonia yield rate were 96.5% and 77 $\mu$mol h$^{-1}$ cm$^{-2}$, with almost no $NO_2^-$ in the products (Figure 4c). In addition to the colorimetric methods, the concentration of products was verified using ion chromatography (Figures S7–S9). The calculated concentrations from ion chromatography were almost the same as the colorimetric method (Table S3). As the reaction proceeds, the concentration of $NH_4^+$ gradually increased, while the concentration of $NO_3^-$ decreased, suggesting the reaction of $NO_3^- \rightarrow NH_4^+$ (Figure S10). After 5 h of reaction, the conversion rate of $NO_3^-$ became flat, and the growth rate of $NH_4^+$ also decreased, implying that the reaction on the surface of CuBMMB reached equilibrium. The results of time-dependent product contents showed that more than 70% of $NO_3^-$ had converted to $NH_4^+$ and $NO_2^-$, where the $NH_3$ selectivity was 96.5%. The catalytic performance of CuBMMB at different concentrations (50, 100, 200, and 400 ppm) was also investigated, and the LSV curves for each condition were shown in Figure 4d. There is no doubt that the current density increased faster as the nitrate concentration increased. As shown in Figure 4e, the FE of $NH_4^+$ production reached a maximum value (97%) at 100 ppm $NO_3^-$-N, and the 400 ppm $NO_3^-$-N reached the minimum value (71.7%). Moreover, the impact on selectivity of $NH_3$ production was studied with all $NO_3^-$-N concentrations, and the values reached the maximum value (96.5%) at 200 ppm $NO_3^-$-N. Stability is a crucial factor for electrocatalysts in practical applications. As presented in Figure 4f, FE and $NO_3^-$ conversion rate showed a significant decrease after three consecutive cycles (each cycle lasts 5 h). Due to the coordination of Cu with nitrate, which leads to an unstable material surface during the electrochemical process, the material surface inevitably reconfigures as the reaction time increases, which leads to the exposure of more active sites and acceleration of the hydrogen evolution reaction; therefore, further destroying the crystal structure of the complex. The number of active sites has an important influence on the activity of electrocatalyst. As a result, electrochemical active surface areas (ECSA) were derived from double-layer capacitances (Cdl), which was determined by the functional relationship between capacitive currents and scan rates (Figure S11). The results of linear fitting indicated that the specific capacitance of CuBMMB was 0.28 mF cm$^{-2}$. Tafel plot obtained from the LSV curves in 1 M KOH electrolyte with 200 ppm $NO_3^-$-N was shown in Figure S12, with CuBMMB exhibiting a modest Tafel slope of 228.9 mV dec$^{-1}$.

Characterization of the composition and structure of the CuBMMB after the NRA were conducted by SEM, XRD, and XPS in Figure 5. According to the SEM image (Figure 5a) of the CuBMMB after the reaction, the surface of the material was clearly reconstructed. The small, clear holes on the surface before the reaction have turned into craters, and the spread-out layers have gathered into blocks. Moreover, the XRD image (Figure 5b) proved that the crystal structure of CuBMMB was destroyed. The FE and $NO_3^-$ conversion rate dropped significantly after three cycles, which was mainly due to the competing HER. By comparing Figure 2c with Figure 5a, we can infer that the disruption of the

CuBMMB structure exposes a large number of active sites and breaks the equilibrium between the HER and NRA reactions, which significantly facilitates the HER reaction. Therefore, an increasing number of bubbles exist on the material surface with the reaction. With regard to the high-resolution XPS spectra of Cu 2p (Figure 5c), compared to the initial complex, the Cu $2p_{1/2}$ spectra exhibit a slight negative shift, demonstrating that the copper ions in the complex become more stable after NRA, with an increase in the content of $Cu^{2+}$ (50.22 → 55.48%) [45]. However, two peaks (292.5 and 295.5 eV) at high binding energies in the spectra of C 1s after NRA (Figure S13) indicate a disruption in the material structure, which is consistent with the XRD and SEM characterization findings. The rate-determining step reaction ($NO_3^-{}_{(ad)} \rightarrow NO_2^-{}_{(ad)}$) first adsorbs $NO_3^-{}_{(aq)}$ on the surface of the electrocatalytic material to form $NO_3^-{}_{(ad)}$, and then obtains electrons to generate $NO_2^-{}_{(ad)}$ [18]. The electron transfer step is very fast, and the key point is the adsorption of $NO_3^-$. Notably, benefiting from the straightforward and efficient design approach, the adsorption capacity of Cu ions for $NO_3^-$ is greatly increased, thus accelerating the rate-determining step $NO_3^- + H_2O + 2e^- \rightarrow NO_2^- + 2OH^-$ and converting almost all of the $NO_2^-$ adsorbed to $NH_4^+$ ($NO_2^- + 8H^+ + 6e^- \rightarrow NH_4^+ + 2H_2O$) (Figure 5d). The electrocatalytic activity of CuBMMB (initial coordination polymer) is comparable to the MOF-loaded metal particle catalysts (Table S4).

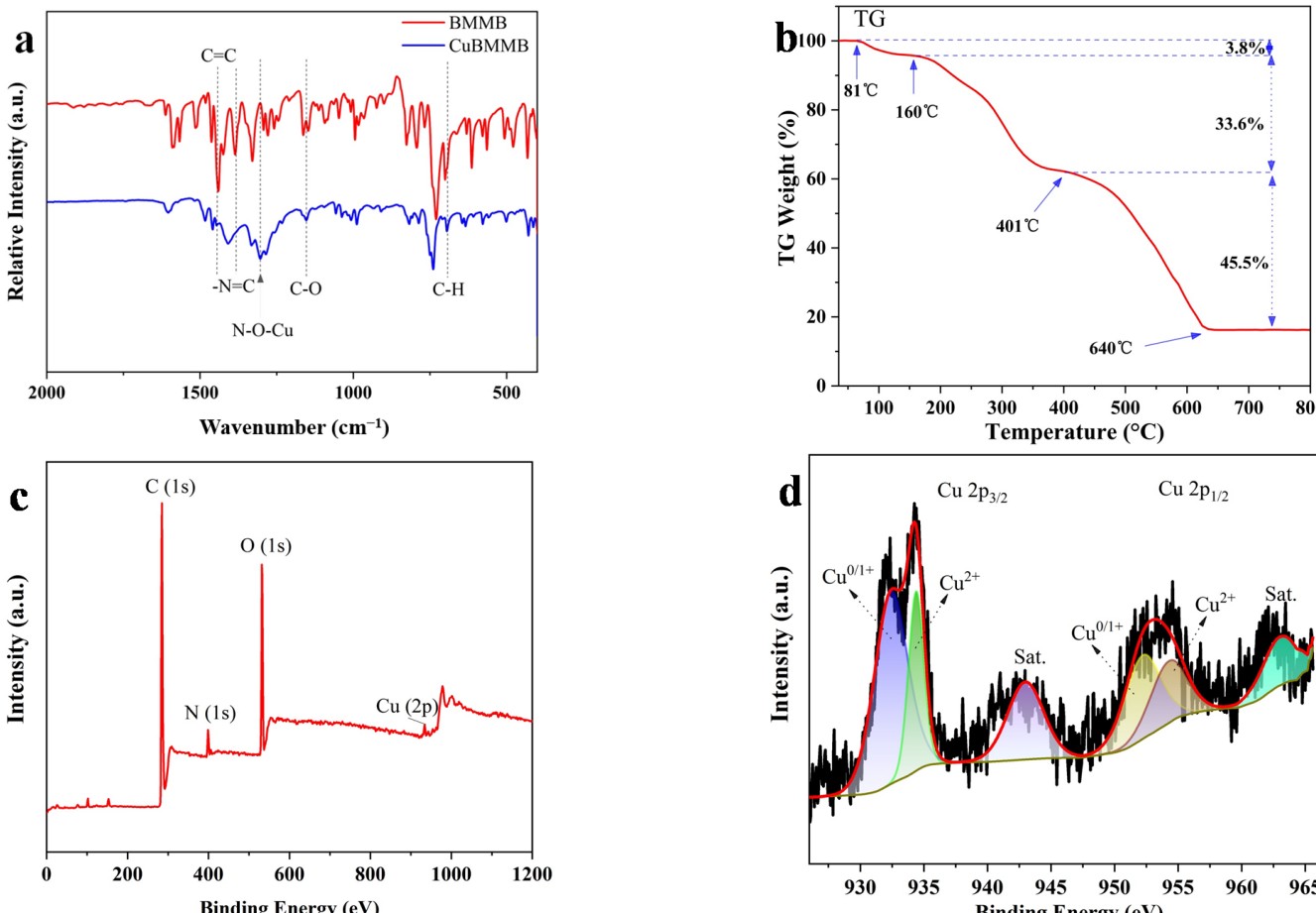

**Figure 3.** (**a**) FT-IR; (**b**) TG spectra of CuBMMB; and XPS survey of (**c**) CuBMMB and (**d**) Cu 2p.

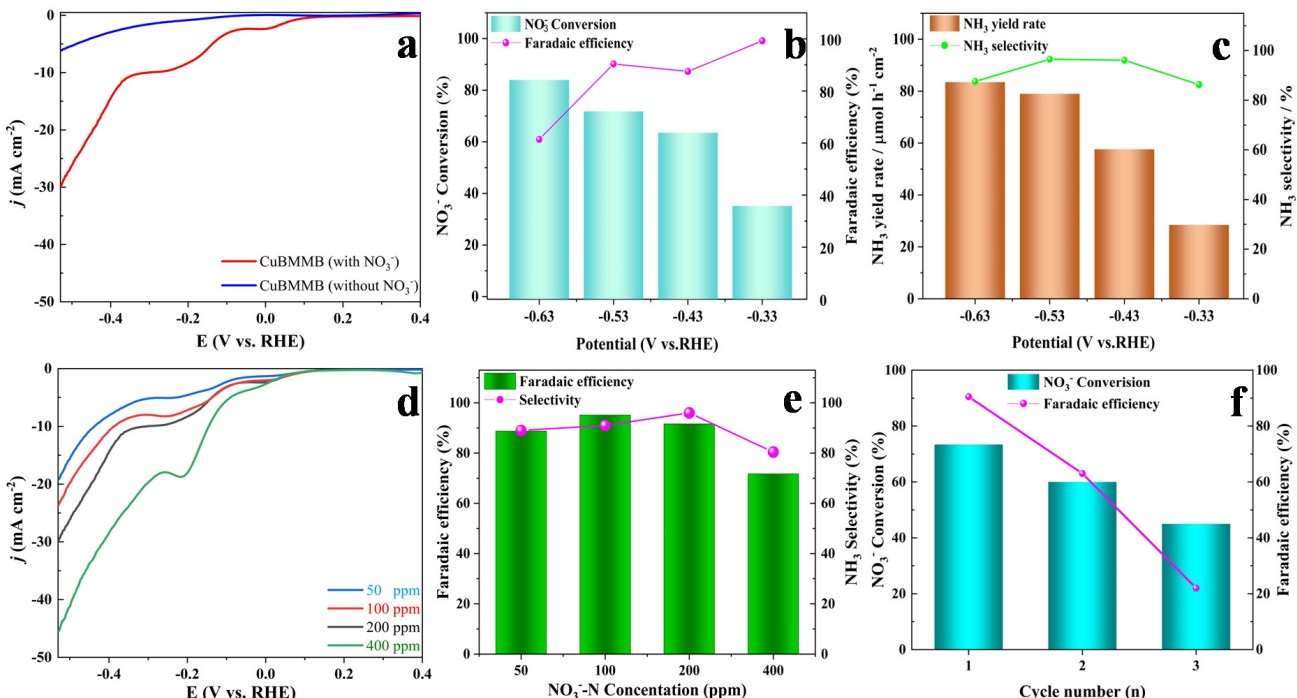

**Figure 4.** NRA electrocatalytic performance. (**a**) LSV curves of CuBMMB in 1 M KOH electrolyte with and without $NO_3^-$, respectively; (**b**) potential−dependent FE and selectivity of $NH_3$ over CuBMMB in 1 M KOH electrolyte with 200 ppm $NO_3^-$; (**c**) $NH_3$ yield rate and $NH_3$ selectivity of CuBMMB electrodes under different potentials; (**d**) LSV curves of CuBMMB tested with different $NO_3^-$-N concentrations; (**e**) FE and selectivity of $NH_3$ tested at −0.53 V vs. RHE over CuBMMB in 1 M KOH electrolyte with 50, 100, 200, and 400 ppm $NO_3^-$-N concentrations; (**f**) the consecutive recycling test of CuBMMB at −0.53 V vs. RHE 1 M KOH electrolyte with 200 ppm $NO_3^-$.

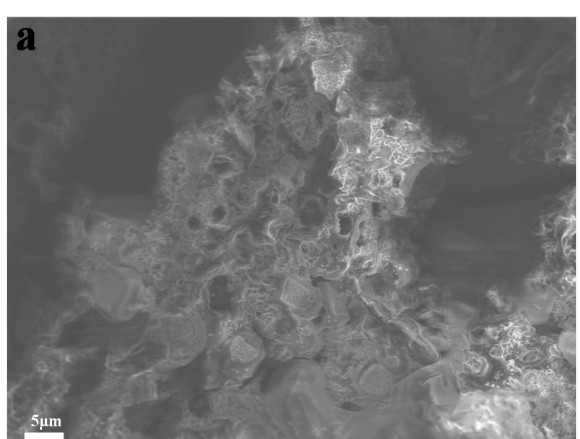

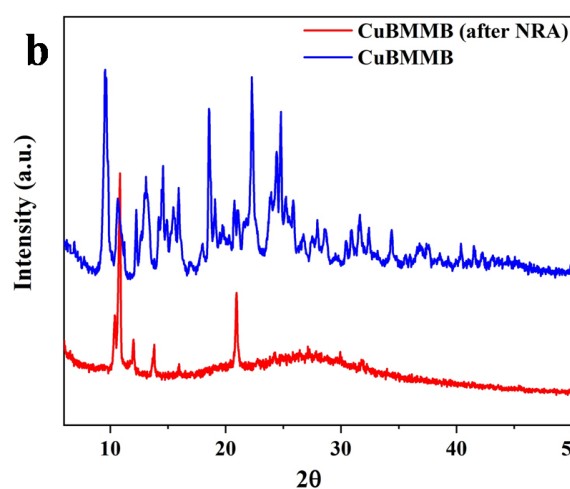

**Figure 5.** *Cont.*

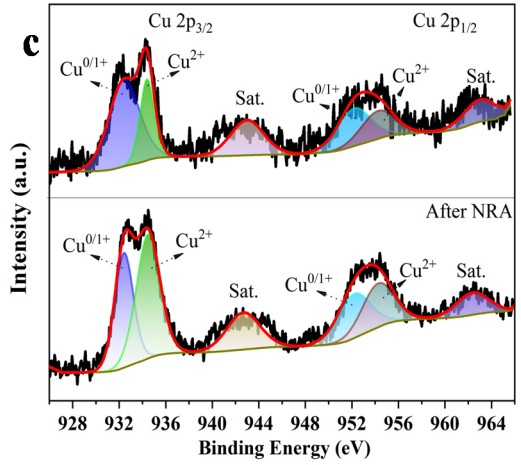
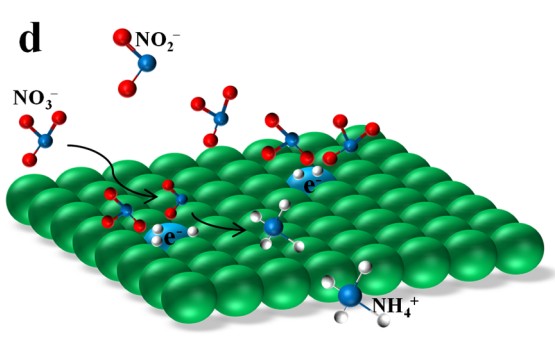

**Figure 5.** Structure, morphology, and composition analysis after NRA; (**a**) SEM; (**b**) XRD; (**c**) and XPS spectra of CuBMMB; (**d**) proposed electrochemical process involved in the nitrate reduction in CuBMMB.

## 3. Materials and Methods

### 3.1. Synthesis of CuBMMB Crystals and Powder

All solvents and reagents for the synthesis were commercially available and used as received. The CuBMMB crystals were synthesized by the solvent-diffusing method. First, 6 mL methanol ($CH_3OH$) solution of $Cu(NO_3)_2 \cdot xH_2O$ (0.1 mmol) was added dropwise at the bottom of a test tube. Then, the solution of $CH_3OH$ and $CHCl_3$ (1:1.8 mL) was carefully added as a buffer layer. Finally, a solution of 1,4-bis{[2-(2'-pyridyl)benzimidazolyl]methyl}benzene (BMMB) in $CHCl_3$ (6 mL) was layered onto the buffer layer, and the test tube was then arranged at an undisturbed place. After ~3 weeks, green single crystals appeared on the wall of the test tube.

The Cu-MOF powder was synthesized as follows. The ligand BMMB and $Cu(NO_3)_2 \cdot xH_2O$ were dissolved in $CHCl_3$ and $CH_3OH$ to obtain a clear solution, respectively. Subsequently, the two clear liquids were mixed and stirred at room temperature for 12 h. Then, the green powder precipitated and thereafter was centrifuged and rinsed with deionized water and ethanol several times. Finally, the powder was evacuated at 60 °C under vacuum for 6 h.

### 3.2. Characterization

X-ray diffraction (XRD) was recorded with Bruker D8 Advance powder XRD system (Billerica, MA, USA) using Cu Kα radiation (λ = 1.54 Å). IR spectra were recorded on Thermo-Nicolet 6700 (Thermo Fischer Scientific, Kandel, Germany) with air as a reference. Field emission scanning electron microscopy (FE-SEM, Hitachi SU8010, Tokyo, Japan) has been utilized to obtain morphology information on the samples. X-ray photoelectron spectroscopy (XPS) survey was performed on Thermo Fisher ESCALAB Xi+ instrument and the non-monochromatized Al-Ka X-ray was used. Thermogravimetry analysis was conducted on METTERLER under air atmosphere with the heating rate of 5 °C min$^{-1}$ in the range of 35–800 °C.

### 3.3. Electrochemical Measurements

Electrochemical measurements were conducted on a CHI 660E electrochemical workstation (Chinstruments, Shanghai, China) using an H-type electrolytic cell separated by a membrane (Nafion-117). Saturated calomel electrode (SCE), Pt foil electrode, and as-prepared electrode were used as reference electrode, counter electrode, and working electrode, respectively. The 1 M KOH solution (40 mL) with different concentrations of $KNO_3$ was used in the cathode (containing 200 ppm nitrate-N) and anode (containing 0 ppm nitrate-N) compartments. Linear sweep voltammetry (LSV) was performed at a scan rate of

10 mV s$^{-1}$ to obtain polarization curves from 0.40 to $-0.53$ V (vs. RHE). The potentiostatic curve tests were performed at different potentials ($-0.33$, $-0.43$, $-0.53$, and $-0.63$ V vs. RHE) for 5 h with a stirring rate of 300 rpm. The electrochemical double-layer capacitance ($C_{dl}$) using the cyclic voltammetry (CV) method was nearly a non-Faradaic potential window (from 0.97 to 1.07 V) at different scan rates of 10, 20, 40, 60, 80, 100 mV s$^{-1}$. The concentrations of nitrate-N, nitrite-N, and ammonia-N were detected by UV–Vis spectrophotometry (Shimadzu UV-2600i) according to the standard method. Meanwhile, the concentrations of ammonia-N were also confirmed by ion chromatography (Shine CIC-D100). The detailed determination methods are provided in the Supporting Information (SI).

## 4. Conclusions

In summary, the coordination polymer CuBMMB proved to be a novel NRA electrocatalyst with outstanding activity, good nitrate conversion rate, high NH$_3$ selectivity, and FE of NH$_3$. Under the desirable potential of $-0.53$ V vs. RHE, CuBMMB in alkaline conditions showed excellent nitrate reduction performance with the NO$_3^-$ conversion rate of 70%, Faradaic efficiency of up to 90%, and NH$_3$ selectivity of more than 95%. This high-catalytic activity is mainly due to the ingeniously designed copper cores with strong affinity for NO$_3^-$, which accelerates the transferring rate of adsorbed nitrate on the Cu surface and increases the efficiency of rate-determining step (NO$_3^- \rightarrow$ NO$_2^-$) for the whole catalytic process. Therefore, the transformation of surface-exposed nitrate can be rapidly catalyzed by the Cu active sites, and the high affinity of Cu ion for nitrate greatly promotes the efficiency of nitrate conversion.

**Supplementary Materials:** The following supporting information can be downloaded at: https://www.mdpi.com/article/10.3390/catal12121561/s1, Figure S1: SEM image of the CuBMMB; Figure S2: The EDS images of the CuBMMB; Figure S3: The XPS survey of C 1s, N 1s, and O 1s; Figure S4: Concentration-absorbance calibration curves of (a) ammonia-N, (b) nitrate-N, and (c) nitrite-N; Figure S5: LSV curves in 1 M KOH electrolyte with 200 ppm NO$_3^-$-N; Figure S6: Faradaic efficiency and nitrate conversion of the CuBMMB and carbon cloth in 1 M KOH electrolyte with 200 ppm NO$_3^-$-N at $-0.53$ V (vs. RHE); Figure S7: Ion chromatogram (IC) curves of the time-dependent ammonium production for a series of standard ammonium solutions and CuBMMB at $-0.53$ V; (b) calibration curve of IC for ammonium concentration; Figure S8: IC curves of the time-dependent nitrate residue for a series of standard nitrate solutions and CuBMMB at $-0.53$ V; (b) calibration curve of IC for nitrate concentration; Figure S9: IC curves of the time-dependent nitrite production for a series of standard nitrite solutions and CuBMMB at $-0.53$ V; (b) calibration curve of IC for nitrite concentration; Figure S10: Time-dependent concentration of NO$_3^-$, NO$_2^-$, and NH$_4^+$ over CuBMMB at $-0.53$ V vs. RHE; Figure S11: (a) CV curves of the CuBMMB at different scanning rates (10–100 mV s$^{-1}$); (b) relationship between the scanning rate and the current density of Cu-MOF; Figure S12: Tafel plot of CuBMMB in 1 M KOH electrolyte with 200 ppm NO$_3^-$-N; Figure S13: The XPS survey of C 1s, N 1s, and O 1s after NRA; Table S1: The element distribution of CuBMMB; Table S2: The value of total nitrogen element at different times; Table S3: The value of ion chromatography and colorimetric methods for the concentration of produced nitrite and ammonium as well as residual nitrate; Table S4: Performance comparison; References [3,5,24,25,29,30,34–36] are cited in Supplementary Materials.

**Author Contributions:** Z.-X.L. conceived the idea, discussed results and revised the manuscript. J.-J.W. analysed crystal data and revised the manuscript. K.-Y.Z. tested electrochemical data, discussed results and prepared the manuscript. X.F. performed syntheses. All authors have read and agreed to the published version of the manuscript.

**Funding:** This work was supported by the National Natural Science Foundation of China (21673177, 22171205).

**Data Availability Statement:** Data openly available in a public repository, which can be free downloaded in the website of mdpi system.

**Acknowledgments:** K.Y.Z. thanks Feng Du for his assistance in the measurements and discussion of NRA.

**Conflicts of Interest:** The authors declare no conflict of interest.

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
