# Peer review of "One Bicopper Complex with Good Affinity to Nitrate for Highly Selective Electrocatalytic Nitrate Reduction to Ammonia"

_catalysts, doi:10.3390/catal12121561_

Round 1

Reviewer 1 Report

1- The title needs modification to reflect the actual catalyst composition

2- In the introduction Full names of the process/compound abbreviations should be given such as NRA, CuHHTP, CuBMMB, Th-BPYDC, etc…

3- Higher magnification SEM images with better resolution of the CuBMMB are mandatorily required to identify the proper morphology. The current SEM images are not conclusive

4- Also, SEM elemental mapping of Cu, N and O is completely black, better images must be provided.

5- Identify the diffraction planes on XRD of figure 2a

6- In LSV of CuBMMB in Fig. 4a, 4d, and Fig. S5 explain the origin of the waves around the potential of 0.0 and -0.2 V. Is the electrolyte contains dissolved oxygen? That could compete with nitrate reduction reaction.

7- The catalytic performance of CuBMMB and the ammonia production rate should be compared with the literature of relevant catalysts  

Reviewer 2 Report

In the manuscript catalysts-2001961 with the title “One bicopper complex with good affinity to nitrate for highly selective electrocatalytic nitrate reduction to ammonia”, authors reported a novel nitrate to ammonia/ammonium (NH4+) electrocatalyst with high Faradaic efficiency (FE, 90%) to NH4+ and high nitrate conversion rate (70%). A Cu-based complex with nitrate and 1,4-bis{[2-(2-pyridyl)benzimidazolyl]methyl}benzene (bmmb) was synthesized, aiming to promote nitrate to ammonia (NRA) conversion on the Cu site by improving the adsorption of nitrate to the Cu active site. Material characterization including XRD, SEM/EDX, and XPS were conducted on the pristine and post-mortem electrode materials. However, authors failed to precisely and accurately interpret the characterization results. Conclusions were draw based on speculation rather than concrete evidence. Furthermore, the manuscript have numerous language issues, i.e. non-scientific, vague language, non-defined abbreviation. Therefore, I judged this manuscript is not yet suitable for publication in this journal. My detailed reviews are given as follows:

1)      There are several key conclusions drew based on mainly speculation rather than experimental evidence:

a.       Line 240 – 243: authors claim that adsorption capacity of NO3- on Cu was increased by their design strategy but no evidence was provided to support such claim; the relationship between the increased adsorption capacity of NO3- on Cu and the increased rate-determining reaction rate is largely speculated, and lack of explanation.

b.       Line 214 – 217: authors relate the decreased conversion rate and FE to the increased active sites due to the reconstructed complex during the electroreduction. Such claim is vague, unchecked and speculative. Firstly, the decrease of FE to NH3 is not necessary due to HER but also possibility NO3- to NOx and/or N2 conversion. Furthermore, there is lack of evidence to show that the reconstruction of complex leads to increasing of the active site. At last, no explanation was given to explain why accelerated HER can destroy the crystal structure.

2)      Some of the results were also improperly interpreted:

a.       Line 118 – 119: authors claim that the XRD spectra of the synthesized CuBMMB material matches well with the simulated standard pattern. However, there are numerous peaks cannot be indexed by the CuBMMB standard XRD pattern, such as extra peaks at 2θ = 10.5-12°, 13.5-14.5°, 17-18°, etc.

b.       Line 130 – 131: the FT-IR spectrum of BMMB and CuBMMB are shown and authors claim that the two spectrums are very similar. Such interpretation of the data is not specific and did not clearly demonstrate the purity of the CuBMMB.

c.       Line 136 – 139: authors reported the TG curve of the material. However, it did not supply any useful information to the discussion of the catalytic activity/selectivity of the material, nor has it provided information to explain the poor durability of the CuBMMB catalyst. Meanwhile, there is a large mismatch between the calculated (17%) and experimentally obtained (11.3%) residual CuO mass, which could be interpreted as an evidence to show that the synthesized material is in fact not pure.

d.       Line 184 – 185: authors concluded that the carbon cloth contribute negligibly to the NO3- reduction. However, the conversion rate of NO3- due to carbon cloth was found around 7%, which is approximately 1/10 of the total conversion rate of NO3- when coated with the CuBMMB. This is not a negligible value.

e.       Line 229 – 230: authors claim that crystal structure of CuBMMB was maintained after the electroreduction. On the contrary, the post-mortem XRD results is vastly different from that of the pristine CuBMMB.

f.        In supplementary information: in equation 3, the calculation of NH3 selectivity is done by dividing the change of NH3 concentration with the change of NO3- concentration. As explained by the authors, the NO3- bonded to the Cu in the complex can also be reduced hence the estimation of the NH3 selectivity will always be estimated because the amount of coordinated NO3- consumed during the electroreduction process is never taken into account.

3)      Non-specified abbreviation

a.       Line 65 – 73, [Co(DIM)]3+, Th-BPYDC, RHE, CuHHTP, CeBDC.

b.       Line 139: XPS

c.       Line 225: S1+Cu

d.       Line 253: S1

4)      Poorly used words and constructed sentences, e.g.,  

a.       Line 38:  denitrification à denitrify

b.       Line 43: above à those mentioned above

c.       Line 54: very good, not scientific language

d.       Line 96: transferring rate of nitrate adsorption à transferring rate of adsorbed nitrate

e.       Line 88 – 90: Firstly, the reason semirigid organic … is chosen as …

f.        Line 231: blamed à due

g.       Line 265: One utilize field emission à field emission .. has been utilized

5)      Grammar issues, e.g.,

a.       Line 33: nitrate can be converted

b.       Line 35: In addition, high concentration

c.       Line 70: single-site

d.       Line 77: improving

e.       Line 235: the Cu 2p1/2 exhibit a slight negatively shifted. Demonstrating à the Cu 2p1/2  spectra exhibit a slight negatively shifted, demonstrating the

f.        Line 267: survey was performed

Reviewer 3 Report

Dear Editor, it is delighted to have the opportunity to review the proposed paper (manuscript ID: catalysts-2001961) in the MDPI Catalysts Journal.

The present manuscript has made interesting findings to scientific community. Briefly, this study reported a conversion of nitrate pollutants to ammonia by using an electrochemical reduction technique. The subject is interesting, but it needs revisions to promote the manuscript quality and interest for the readers of the Catalysts journal. The points are as follow:

1.     The authors needs to check the language of the whole manuscript, as there are many small mistakes for example some time they used Figure, and Fig. …..etc.

2.     The resolution of all figures needs to be enhanced (specialty Figure 4)

3.     The authors need to discuss in more details the physical characterization (XRD, FTIR, TGA, SEM, XPS) of the prepared complex as in most cases they just add a small detail, which is not sufficient to the readers.

4.     For the EDS, the authors need to add the findings in the main manuscript not just say in supplementary data

5.     The authors mention that the CuBMMB is an effective in the NRA, can they mention the estimated parameters that provide their claims?

Regards

Round 2

Reviewer 1 Report

The author successfully revised the manuscript as recommended.

The work now can be accepted for publication. 

Author Response

Thank you for your approval.

Reviewer 2 Report

In the revised manuscript (ID: catalysts-2001961) with title “One bicopper complex with good affinity to nitrate for highly selective 2 electrocatalytic nitrate reduction to ammonia”, authors improved the language and corrected some misinterpretation of experimental data. However, there are still a few issues needs to be addressed:

1)      Authors tried to use Figure 2a to illustrate that the CuBMMB synthesized was pure and well-crystallized. However, there are still unidentified peaks at e.g. 11, 13.5, 16.5 deg, which could be attributed to sample holder or impurities. Furthermore, the measured XRD diffractogram shows a clear displacement of the sample during measurement, causing peaks shift to a higher 2θ angle in comparison to the simulated one.

2)      In Figure 4b and 4c, the author showed that when working electrode potential is at -0.33 V vs RHE, the Faradaic efficiency to NH4+ is nearly 100%. However, the selectivity to NH3 is merely 82%, suggesting that there are more NO3- consumed than NH4+ produced. This cannot happen if authors assume that there is only electrochemical conversion of NO3- to other N-containing species.

3)      Eq. 2 and 3 in the supplementary information: Given that CuBMMB contains extra NO3- and those extra NO3- are consumed during the electrochemical reduction reaction, the change of NO3- concentration (later the calculated NO3- conversion rate) in the catholyte is usually underestimated. The NH3 selectivity, on the contrary, is overestimated.

4)      From Figure 5c authors claim, “the compositional of the complex has become more stable after NRA”. Such claim is opposite to what XRD, SEM results suggest.

5)      Post-mortem XRD, SEM results suggest there is no CuBMMB left after the reduction. Authors can try to identify which Cu-containing composition can be indexed to the peaks in Figure 5b.

6)      There is no mentioning of the instability of the catalyst in the conclusion, given that the selectivity decreased over 70% after 3 circles.
